# Intra Prediction Method for Depth Video Coding by Block Clustering through Deep Learning

**DOI:** 10.3390/s22249656

**Published:** 2022-12-09

**Authors:** Dong-seok Lee, Soon-kak Kwon

**Affiliations:** 1AI Grand ICT Research Center, Dong-eui University, Busan 47340, Republic of Korea; 2Department of Computer Software Engineering, Dong-eui University, Busan 47340, Republic of Korea

**Keywords:** intra prediction, depth video coding, deep learning, 1D CNN, clustering

## Abstract

In this paper, we propose an intra-picture prediction method for depth video by a block clustering through a neural network. The proposed method solves a problem that the block that has two or more clusters drops the prediction performance of the intra prediction for depth video. The proposed neural network consists of both a spatial feature prediction network and a clustering network. The spatial feature prediction network utilizes spatial features in vertical and horizontal directions. The network contains a 1D CNN layer and a fully connected layer. The 1D CNN layer extracts the spatial features for a vertical direction and a horizontal direction from a top block and a left block of the reference pixels, respectively. 1D CNN is designed to handle time-series data, but it can also be applied to find the spatial features by regarding a pixel order in a certain direction as a timestamp. The fully connected layer predicts the spatial features of the block to be coded through the extracted features. The clustering network finds clusters from the spatial features which are the outputs of the spatial feature prediction network. The network consists of 4 CNN layers. The first 3 CNN layers combine two spatial features in the vertical and horizontal directions. The last layer outputs the probabilities that pixels belong to the clusters. The pixels of the block are predicted by the representative values of the clusters that are the average of the reference pixels belonging to the clusters. For the intra prediction for various block sizes, the block is scaled to the size of the network input. The prediction result through the proposed network is scaled back to the original size. In network training, the mean square error is used as a loss function between the original block and the predicted block. A penalty for output values far from both ends is introduced to the loss function for clear network clustering. In the simulation results, the bit rate is saved by up to 12.45% under the same distortion condition compared with the latest video coding standard.

## 1. Introduction

Depth video stores the distances in pixels which are measured by a ToF (Time of Flight) sensor or are calculated by comparing two pictures captured by stereo cameras. 3D surfaces can be calculated through 2D coordinates and depth pixel values. The applications of depth video through the information of the 3D surface are as follows: the human action is recognized by detecting a body and analyzing a 3D movement from depth video [1,2]; in autonomous driving, an object approaching the driving car is detected by depth videos captured by side-mounted lidar sensors [3,4]; in the immersive video, a video at a specific viewpoint is restored based on texture and distance information [5]. Therefore, depth video compression is required to store and transmit depth video.

The most common method of video compression is a residual coding by a block-based prediction. The block-based method predicts pixels in a rectangular block by referring to temporally or spatially adjacent pixels. The block-based prediction applies to video coding standards such as HEVC (High Efficiency Video Coding) [6] and VVC (Versatile Video Coding) [7]. Even though the intra prediction for the conventional standard algorithms also can be applied for depth video, a coding performance is lower than for color video, because the spatial redundancy for depth video is different from that for color video. The spatial redundancy for color video means the similarity of adjacent pixels, while the spatial redundancy for depth video means the similarity of surfaces. Our previous study [8] improved the performance of the block-based intra prediction through a plane modeling method, which predicts pixels as a point on a plane surface modeled from the reference pixels. Another factor of the low performance of the depth video coding is hard to predict a block which contains two or more clusters. The angular intra prediction modes in the conventional standards can solve this problem a little if a boundary between the clusters is consistently oriented. However, the modes still inaccurately predict the block with non-linear boundaries. The problem greatly reduces the coding performance for depth video due to the extended range of pixel presentation than color video.

Though RNN (Recurrent Neural Network) has been mainly for analyzing time-series data, RNN is also possible to predict the spatial features for the intra prediction by regarding the order of the reference pixels in a vertical or horizontal direction as time [9,10]. However, a recursive structure in RNN causes gradient vanishing or exploding in training. In the proposed network, 1D CNN (Convolution Neural Network) replaces RNN for the spatial feature prediction. Similar to RNN, 1D CNN also takes continuous data to predict features. The structure of 1D CNN is simpler than RNN, so then the network training is easier. The network performance is reported to be better for 3D CNN than for RNN [11,12].

In this paper, we propose an intra prediction method through a neural network for clustering. The network consists of a spatial feature prediction network and a clustering network. The spatial feature prediction network finds spatial features in the vertical and horizontal directions from the reference pixels through 1D CNN. The clustering network calculates the probability that pixels belong to each cluster by CNNs. The pixels are predicted through the cluster prediction result.

This paper is organized as follows. Related works are described in Section 2 about the intra prediction through the neural networks and the depth video compression. In Section 3, we propose the neural network for a block clustering and the intra prediction method through this network. In Section 4, we present the simulation results to show the improvement of the depth data coding compared with VVC, the latest video coding standard. Finally, we will make a conclusion for this paper in Section 5.

The contributions of this paper are as follows. The proposed method can solve the problem of inaccuracy intra prediction for a block with multiple clusters. 1D CNN layers can replace RNN layers for finding the spatial features of video.

## 2. Related Works

### 2.1. Intra Prediction Methods through Neural Network

The intra-picture prediction methods through the neural network are classified into end-to-end methods, that the entire picture is processed, and block-based methods. Autoencoders [13,14,15,16,17] are the typical end-to-end methods for the picture compression. In the autoencoder, an encoder network reduces the features dimension through CNN or RNN in order to compress the input picture. The pixel-based method [18,19] predicts a pixel by inputting the top and left pixels through CNN. The end-to-end methods are hard to apply to conventional block-based video compression frameworks.

Similar to the intra predictions in conventional video coding standards, the block-based methods predict the pixels of a block by referring to the adjacent pixels that are already coded. The conventional intra prediction generally refers to the adjacent pixels in a single line, whereas the block-based methods utilize the pixels in multiple lines. J. Li [20] and I. Schiopu [21] propose neural networks with fully connected layers and CNN layers, respectively, for the intra prediction. However, the methods have a problem that reference pixels with low spatial similarity are able to participate in predicting a pixel of the block. F. Brand [22] proposes a network with an autoencoder structure. Unlike the end-to-end autoencoder networks, the network takes the reference pixels as input. The output of the network is block restoration information through the reference pixels. The block-based intra prediction is also performed by GAN [Generative Adversarial Networks). GAN generates a block from the reference pixels. In GAN training, the generated pictures are discriminated as to whether they are real together with actual pictures in order to improve a generation performance. In Zhu’s work [23], both two regions of the block to be coded and the reference pixels are discriminated in the network training. G. Zhong [24] proposes a GAN with two-stage coarse-to-fine architecture. A coarse generator generates a coarsely predicted block from the reference pixels. A fine generator generates a more detailed block from the coarse block. The performances of the intra-predictions through GANs have a limitation because pixels at the bottom and right of the block cannot be referenced for generating the predicted block. The intra prediction methods through networks with RNN layers regard the reference pixels in a vertical or horizontal direction as consecutive sequential data. Y. Hu [9] proposes the intra prediction method through RNN layers with various input sizes. The region of the reference pixels is scaled to the input sizes. The PS-RNN (Progressive Spatial Recurrent Neural Network) [10] predicts the visual features of the reference pixels through CNN. The spatial features are extracted through RNN layers guided by the visual features. The pixels in the block are predicted by converting the spatial features to the pixel domain. The neural networks with CNN can improve computational complexities of the block split [25] and the mode selection [26] for the intra prediction. proposes the video quality enhancement method in video decoding by the picture prediction through CNN. Lee et al. [27] propose the video quality enhancement method in video decoding by the picture prediction through CNN.

### 2.2. Depth Video Compression

Depth video has three-dimension spatial information. Therefore, depth video can be converted to a point cloud which is a discrete set of three-dimensional points. MPEG standardizes the point cloud compression as V-PCC (Video-based Point Cloud Compression) [28], which is a patch-based cloud point compression method. After splitting the 3D points in the point cloud into patches with high spatial correlation, each patch is projected onto a 2D surface. The projected patches are compressed through the prediction methods in the conventional video coding standards such as HEVC and VVC. However, the point cloud compression method by 2D projection removes the spatial correlation of Z-axis, so the compression performance has a limit. The point cloud can also be compressed through 3D spatial correlation. Many studies of the point cloud compression generate an octree for dividing 3D space. The octree is a tree structure where nodes represent bounding boxes that are recursively divided into eight leaves. Similar to divided 2D image into blocks through a quadtree, the 3D space can be divided into sub-cubes through octree. The loss and the rate of the compression can be determined by adjusting the depth of the octree. Garcia [29] proposes a method of compressing the flags of leaf nodes and their parents in the octree through LZW algorithm and arithmetic coding. Kathariya [30] proposes a BTQT (Binary Tree Quadtree) structure for compressing the point cloud. The points in the point cloud are split into two sets through a binary tree. A set which presents a plane surface is compressed by converting it into a quadtree. The other is compressed through the octree. The point cloud can also be compressed through voxelization [31]. Adjacent 3D points are converted into a single voxel through voxelization. The point cloud compression methods through 3D spatial correlation are more precise predictions than 2D projection methods. However, these methods have a limitation that has a high computational complexity due to calculating the 3D spatial correlation.

Depth video can also be treated as a single channel video whose range of pixel representation is extended. Therefore, depth video can be compressed by the conventional video coding standards. Nenci [32] proposes a depth video coding method through depth video conversion. The channel of depth video is divided into 8-bit multi-channels. The multi-channel video is compressed through AVC. Wang [33] proposed the inter-frame prediction method of finding camera movements between temporally adjacent depth pictures. Our previous study [8] proposes a new intra prediction mode based on a plane surface estimation. The 3D plane surface is estimated through the reference pixels. The depth pixels are predicted through the estimated plane surface. The study greatly improves the performance especially of intra prediction for the depth pictures with simple backgrounds.

## 3. Intra Prediction Method by Block Clustering through Deep Learning

### 3.1. Spatial Feature Extraction through 1D CNN

In conventional video coding standards such as HEVC and VVC, the spatial features of the block are extracted through the various modes of angular intra predictions. The angular mode predicts pixels as a reference pixel in a certain direction. Figure 1a shows the predictions results of the angular modes in a vertical and horizontal directions. The errors of the predictions are calculated for the angular intra prediction mode and a DC mode, which predicts pixels as the average of the reference pixels. Then, the prediction mode is selected to make the prediction error smallest. The angular intra prediction can extract the spatial features in the linear direction well, but non-linear spatial features can hardly be extracted as shown in Figure 1c.

Though RNN is designed for the continuous data in essence, RNN can extract the spatial features of a video if a pixel order in a certain direction is regarded as a timestamp. RNN extracts the features based on the data order in 1D domain, but the spatial features of the video are in the 2D domain. Therefore, it is necessary to appropriately combine the spatial features in various directions in order to extract the spatial features in the 2D domain. In PS-RNN [10], RNN layers extract two spatial features in the vertical and the horizontal directions. CNN layers combine both of the spatial features and predict the block. Figure 2a shows the extraction of the spatial features through RNN. The spatial feature extraction through RNN can extract non-linear spatial features that conventional angular modes cannot. 1D CNN can replace RNN in the spatial feature extraction of the video similar to the feature extraction of the time series data. 1D CNN can avoid gradient vanishing or exploding in network training, which is the critical problem of RNN. 1D CNN has equal performance to or better performance than RNN [11,12]. The kernel moves the input block, which is the top or the left reference pixels, in one direction and extracts the vertical or horizontal spatial features, respectively. Figure 2b shows the extraction of the vertical spatial features through 1D CNN.

### 3.2. Block Clustering Network

The block clustering network proposed in this paper predicts multiple clusters from an input block based on the depth pixel values. The network consists of both a spatial feature prediction network and a clustering network. The input of the block clustering network is a 32 × 32 block with values in the range [0, 1]. In the input block, areas of a block to be coded and reference pixels are defined as a target area and a reference area, respectively, as shown in Figure 3. The output of the block clustering network is a 32 × 32 × *C* cluster probability map, where *C* is the number of the clusters to predict. Figure 4 shows the structure of the block clustering network.

#### 3.2.1. Spatial Feature Prediction Network

A spatial feature prediction network finds the spatial features of the input block. Horizontal and vertical spatial features are predicted from left reference and top reference blocks, respectively. The prediction of the horizontal features of the left reference block can be regarded as equal to predict the vertical features of the transposed block. Therefore, the prediction of two spatial features can be performed through the same network. It makes the complexity lower and the learning efficiency improve for the network. Figure 5 shows the transformation of the left reference block.

The spatial features of the reference pixels are extracted through a 1D CNN layer. The kernel size of the 1D CNN layer is 16 × 3, which is the same height as the input block. Then, the spatial features in the pixel unit are predicted through a fully connected layer. PReLUs (Parametric Rectified Linear Unit) [34] are used as activation functions in the block clustering network. PReLU outputs the input value directly if the input value is greater than 0, otherwise outputting the input value multiplied by a learned parameter. The output vector is reshaped to a 32 × 32 block. Figure 6 shows the structure of the spatial feature prediction network.

The output for the left reference block is transposed again. Then, both the horizontal and the vertical spatial features are concatenated. Figure 7 shows the flow of the feature prediction network. The shared weight of two networks means that both networks are actually identical. Table 1 specifies layers in the network.

#### 3.2.2. Clustering Network

The clustering network predicts the clusters of the input block from the spatial features of both horizontal and vertical directions. The clusters are found through multiple CNN layers. The first CNN layer extracts 12 features combining vertical and horizontal directions. The second CNN layer reduces the dimension of the feature map by a 1 × 1 kernel. The third CNN layer refines the spatial features. The cluster probability for each pixel is calculated by the softmax activation function in the last CNN layer. The output of the clustering network is a 32 × 32 × *C* cluster probability map. The network outputs are cluster probabilities for the input block. The ranges of the output values are [0, 1]. When an output value in the (*i*, *j*, *c*) position is closer to 1, it means a pixel in the (*i*, *j*) position has a higher probability to belong to the *c*th cluster. Figure 8 and Table 2 show the structure and specification of the clustering network, respectively.

### 3.3. Intra Prediction through Block Clustering Network

In order to predict a *M* × *N* block through the proposed network, the 2*M* × 2*N* block including adjacent already coded pixels is inputted. The 2*M* × 2*N* block is scaled to 32 × 32 size by a bilinear interpolation. The pixel values are normalized to the range [0, 1] as follows:(1)$$x\left(i,j\right)=\frac{\left(p\left(i,j\right)-p_{min}\right)}{p_{max}-p_{min}}\;,$$
where *p*(*i*, *j*) and *x*(*i*, *j*) are the values of an original pixel and a normalized pixel in (*i*, *j*) position, respectively, and *p_max_*, *p_min_* are the maximum value and the minimum value of the reference pixels, respectively.

After finding the clusters of the block through the proposed network, the pixel is predicted as a sum of multiplication between the cluster probabilities and the representative values of the clusters in the position of the pixel. The representative value of the *c*th cluster is calculated as the average of the reference pixel values in the *c*th cluster, as follows:(2)$$m_{c}=\frac{\sum_{i,j\in reference}\left(\hat{y}\left(i,j,c\right)\times x\left(i,j\right)\right)}{\sum_{i,j\in reference}\left(\hat{y}\left(i,j,c\right)\right)}\;\;,$$ where *m_c_* is the representative value of the *c*th cluster and $\hat{y}\left(i,j,c\right)$ is the probability of the *c*th cluster in the (*i*, *j*) position. A pixel value $\hat{p}\left(i,j\right)$ is predicted as follows:(3)$$\hat{p}\left(i,j\right)=\overset{C}{\underset{c=1}{\sum }}\left(m_{c}\times \hat{y}\left(i,j,c\right)\right)\;.$$

The 16 × 16 target area of the predicted block is cropped. The area is scaled back to the original size *M* × *N* for the intra prediction. Figure 9 shows the proposed intra prediction through the block clustering network.

### 3.4. Network Training

#### 3.4.1. Dataset for training

The proposed network is trained through 1446 depth pictures about different scenes in NYU Depth Dataset v2 [35]. Figure 10 shows the samples of the depth picture for the network training. Block measuring 16 × 16, 32 × 32, and 64 × 64 at various locations are cropped in the depth pictures. The cropped blocks are clustered into two areas by the K-mean algorithm. The following blocks are not used for the network training: the difference between the pixel averages of two areas is 500 or less; either of two areas occupies less than 30% of either the reference or the target area. The network is trained by about 151,000 blocks in 1157 depth pictures, which is 80% of the total pictures, and is validated through about 51,000 blocks in the others.

#### 3.4.2. Loss Function

In the network training can be considered applying MSE (Mean Square Error) between the original and the reference blocks as a loss function, as follows:(4)$$loss=\frac{1}{32\times 32}\overset{32}{\underset{i=1}{\sum }}\overset{32}{\underset{j=1}{\sum }}\left({\left(\hat{p}\left(i,j\right)-p\left(i,j\right)\right)}^{2}\right)\;.$$

The network trained by Equation (4) finds the clusters for the reference area well, while the cluster prediction for the target area is not clear as shown in Figure 11b. The network does not learn the block clustering but learns to find proportions of the representative values for the intra prediction. In order to solve this problem, a penalty is introduced to the loss function as follows:(5)$$loss=\frac{1}{32\times 32}\overset{32}{\underset{i=1}{\sum }}\overset{32}{\underset{j=1}{\sum }}\left({\left(\hat{p}\left(i,j\right)-p\left(i,j\right)\right)}^{2}\right)+\alpha \times penalty\;,$$
where α is the weight for the penalty. The penalty in Equation (5) is higher as the output value is farther from 0 or 1, as follows:(6)$$penalty=\overset{32}{\underset{i=1}{\sum }}\overset{32}{\underset{j=1}{\sum }}\overset{C}{\underset{c=1}{\sum }}\left(1-4{\left(\hat{y}\left(i,j,c\right)-0.5\right)}^{2}\right)\;.$$

The network trained by Equation (5) predicts the clusters well, as shown in Figure 11c.

## 4. Simulation Results

The intra prediction for depth video through the proposed method is compared with VVC, which is the latest video coding standard. VVC has 67 intra modes including a DC mode, a planar mode, and 65 angular prediction modes. The proposed method is added into the intra mode of VVC. The parameters of network training are as follows: a batch size, the number of epochs, and a learning rate are 128, 150, and 1 × 10^−3^, respectively. The number of the clusters and α in Equation (5) are 3 and 3 × 10^−3^, respectively.

### 4.1. Improvement of Intra Prediction

Figure 12 shows the intra prediction results through the intra modes of VVC and the proposed method. The first row is 32 × 32 input blocks. The areas surrounded by the red rectangle are the target area. The 2nd to 3rd rows and 4th to 5th rows are the intra prediction results through the intra modes of VVC and the proposed method, respectively. When a boundary of the clusters is curved, the intra prediction through VVC is inaccurate, while the proposed method predicts the block more accurately.

### 4.2. Prediction Performances Based on Network Structure

Table 3 shows intra prediction errors based on the number of the clusters in the proposed network. The errors are measured as the MSE averages for the validation set, which are not used for the network training. The prediction performance is better in case of 3 clusters than in case of 2 clusters. The prediction errors with 4 clusters increase rapidly. The network training is difficult for 4 clusters because the block to be coded is rarely divided into four or more clusters.

Table 4 shows performances based on the depths of the fully connected layers in the spatial feature prediction network. The depth increases by adding the fully connected layer with 1024 outputs. We also measure the complexities and the processing time of the network. The complexity of the network is measured as the number of the operations in the network, which is the number of the network parameters. The processing time is averaged by measuring the processing time for a block 100 times. Even though the intra prediction performance improves by adding the fully connected layer, the complexity of the network grows much faster than the increase of the prediction accuracy. It shows that adding the fully connected layers is inefficient.

Table 5 shows intra prediction errors based on the depth of CNN layers in the cluster prediction network. The depth increases by adding the CNN layers whose shape is 32 × 32 × 6. In the results, the prediction performance is significantly improved in case of the structure with 4 CNNs than with 3 CNNs. The prediction performance increases little, even though the depth of the CNN layers is increased to 4 or more. The complexity of the network and the processing time increase less than the cases of increasements of the fully connected layers.

Figure 13 shows the intra prediction errors in the network training during 30 epochs. The penalty weights in Equation (5) are given as 0, 1 × 10^−4^, 5 × 10^−4^, 1 × 10^−3^, and 5 × 10^−3^. The results show that the introduction of the penalty improves the intra prediction performance due to more clearly clustering. The optimal penalty weight is 1 × 10^−3^ with a minimum MSE of 0.0483. The intra prediction performances are almost similar when the penalty weights are 1 × 10^−4^, 5 × 10^−4^, and 1 × 10^−3^. However, the intra prediction performance rather sharply drops if the weight is more than 5 × 10^−3^.

### 4.3. Improvement of Coding Performance

We measure the video coding performance for depth video coding by the proposed method through following depth videos: bedroom and basement in [35]; meeting room, kitchen, and desk in [36]; computer and hat in [37]. The depth videos store the distances to pixels in mm. Figure 14 shows the first frames of the depth videos. The depth videos are coded through VTM (VVC Test Model) [38]. Whole frames in the depth videos are coded only by intra prediction. The range of QPs (Quantization Parameter) are from 0 to 50.

For the encoding error of the depth video, it is appropriate to measure the differences in 3D coordinates [8,39,40,41] instead of the PSNR, which is the video distortion metric for color video. The depth value means a Z-axis coordinate in a 3D coordinate system, so the 2D coordinates (*i*, *j*) with a depth pixel value *p*(*i*, *j*) can be converted to the 3D coordinates as follows:(7)$$\begin{array}{l} X=\frac{i}{f}p\left(i,j\right)\\ Y=\frac{j}{f}p\left(i,j\right)\\ Z=p\left(i,j\right)\;\;\;\;\;\;, \end{array}$$
where *f* is the focus length of the depth camera. The difference in 3D coordinates is calculated as follows:(8)$$\begin{array}{cc} e\left(i,j\right) & ={\left(X\left(i,j\right)-\hat{X}\left(i,j\right)\right)}^{2}+{\left(Y\left(i,j\right)-\hat{Y}\left(i,j\right)\right)}^{2}+{\left(Z\left(i,j\right)-\hat{Z}\left(i,j\right)\right)}^{2}\\ & ={\left[\frac{i}{f}\left(\hat{p}\left(i,j\right)-p\left(i,j\right)\right)\right]}^{2}+{\left[\frac{j}{f}\left(\hat{p}\left(i,j\right)-p\left(i,j\right)\right)\right]}^{2}+{\left(\left(\hat{p}\left(i,j\right)-p\left(i,j\right)\right)\right)}^{2}\\ & =f^{-2}{\left(\hat{p}\left(i,j\right)-p\left(i,j\right)\right)}^{2}\left(i^{2}+j^{2}+f^{2}\right)\;, \end{array}$$
where $X\left(i,j\right),\;Y\left(i,j\right),\;Z\left(i,j\right)$ and $\hat{X}\left(i,j\right),\;\hat{Y}\left(i,j\right),\;\hat{Z}\left(i,j\right)$ are 3D coordinates of an original pixel and a predicted pixel in the (*i*, *j*) position, respectively. In the depth videos for the simulation, *f* is 526.370 mm. RMSE (Root Mean Square Error) is the metric of the coding distortion for depth video. RMSE is calculated as follows:(9)$$\mathrm{RMSE}=\sqrt{\frac{1}{MN}\sum_{i=1}^{M}\sum_{j=1}^{N}{\left(e\left(i,j\right)\right)}^{2}}.$$

The unit of RMSE is equal to the unit of pixels which means distance. Figure 15 shows the comparisons between rate-distortion curves for depth video coding through VVC and the proposed method. In the depth videos of bedroom and basement, the coding performance is more improved. These videos capture many objects, so then, the boundaries are more complex. The proposed method codes the depth videos with many boundaries more efficiently. Table 6 and Table 7 shows the improvements of bit rates and RMSEs through the proposed method and our previous method [8] compared with VVC. Through the proposed method, the RMSEs are reduced up to 6.07% and 5.55% when the bit rates are 500 kbps and 1000 kbps, respectively. The bit rates are improved up to 12.45% and 10.63% when the RMSEs are 10 mm and 15 mm, respectively. The points of the performance improvement are different between the proposed method and our previous method, depth video coding by plane modeling. In the depth videos of basement, bedroom, and desk, the coding performance through the proposed method is better than the previous method since the proposed method predicts these depth videos with complex boundaries more accurately. The previous method is better than the proposed method for coding the depth videos of hat and kitchen, which has few objects with simple background. The previous method performs the surface-based prediction, so then, it greatly improves the bit rates and the distortions for the depth videos with a simple background and small number of objects. On the other hand, the proposed method improves the coding performance for the depth videos with a complex background and many objects.

## 5. Conclusions

In this paper, we proposed the intra prediction method for depth video coding by clustering a block through a neural network. The spatial features of an input block were predicted through the 1D CNN layer. The cluster probabilities were calculated from the predicted spatial features. The pixels in the block were predicted through the cluster probabilities. In the simulation results, the bit rates and RMSEs were improved up to 12.45% and 8.04%, respectively. The proposed method can solve the problem of depth video coding whereby it is hard to predict the block divided into multiple areas based on pixel values.

## Figures and Tables

**Figure 1 sensors-22-09656-f001:**
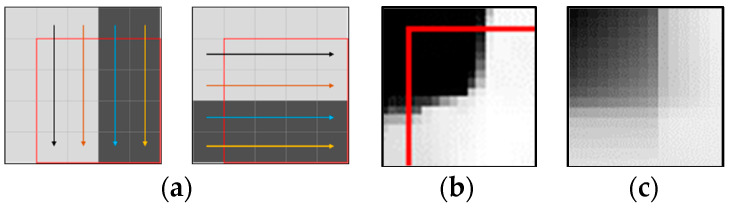
Spatial feature extraction in video coding standards: (**a**) angular intra predictions in a vertical direction and a horizontal direction; (**b**) a block with non-linear spatial features; (**c**) the result of the intra prediction of VVC for (**b**).

**Figure 2 sensors-22-09656-f002:**
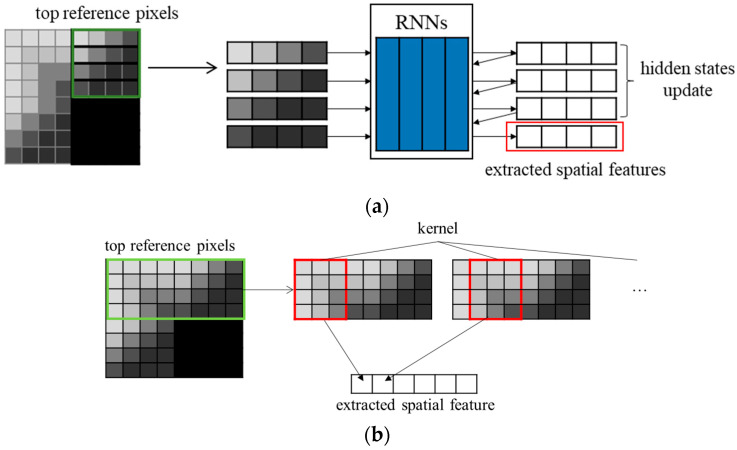
Spatial feature extraction in a block: (**a**) through RNN; (**b**) through 1D CNN.

**Figure 3 sensors-22-09656-f003:**
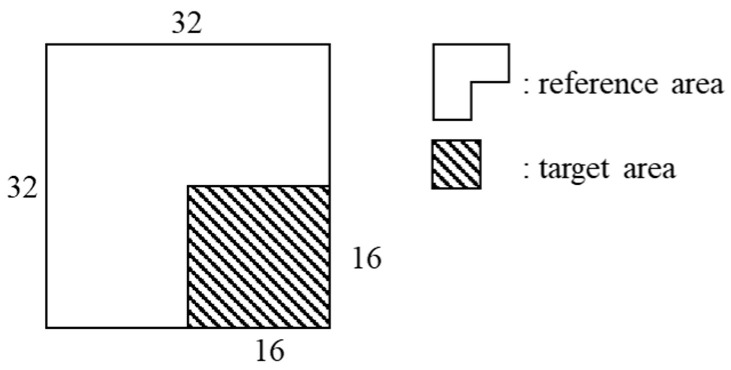
Input block of the proposed network.

**Figure 4 sensors-22-09656-f004:**
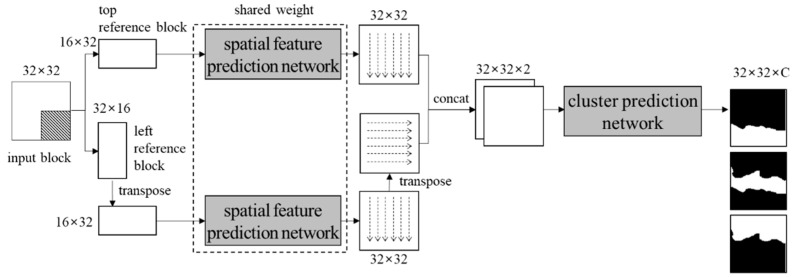
Structure of the proposed network.

**Figure 5 sensors-22-09656-f005:**
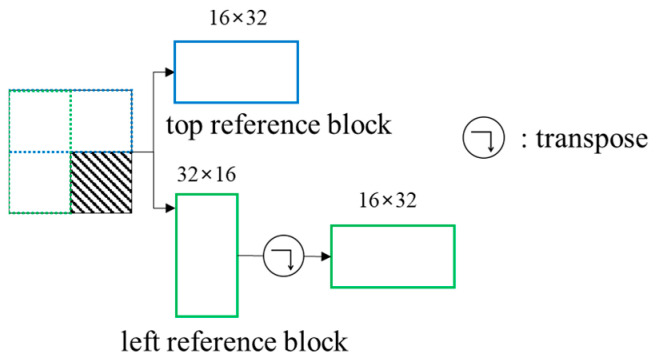
Input of the spatial feature prediction network.

**Figure 6 sensors-22-09656-f006:**
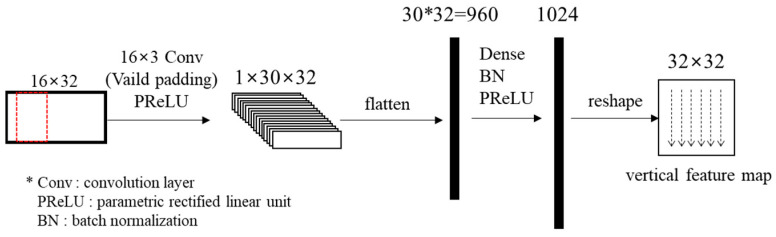
Structure of the spatial feature prediction network.

**Figure 7 sensors-22-09656-f007:**
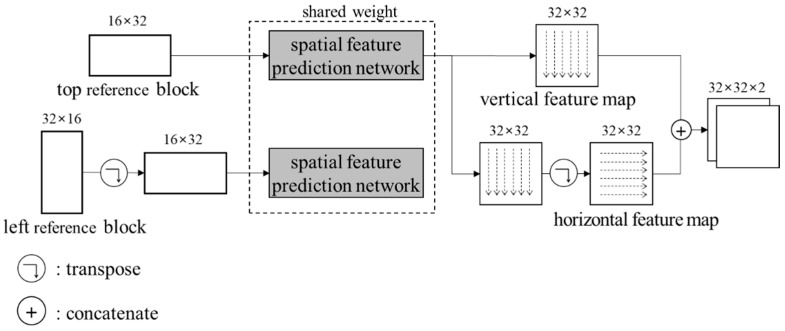
Flow of the spatial feature prediction network.

**Figure 8 sensors-22-09656-f008:**
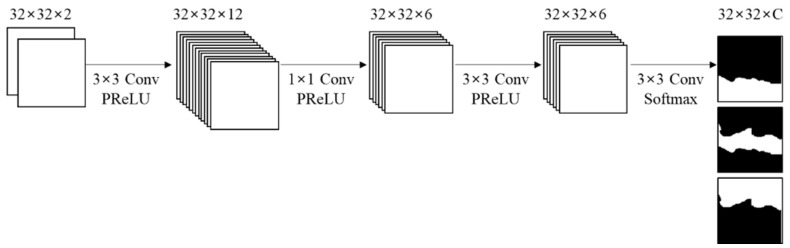
Structure of the clustering network.

**Figure 9 sensors-22-09656-f009:**
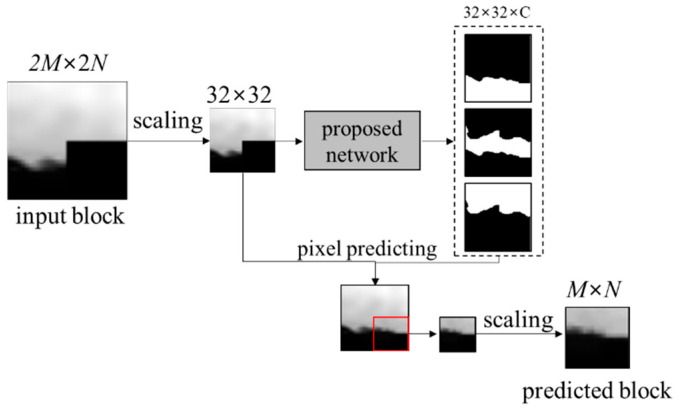
Flow of the proposed intra prediction.

**Figure 10 sensors-22-09656-f010:**
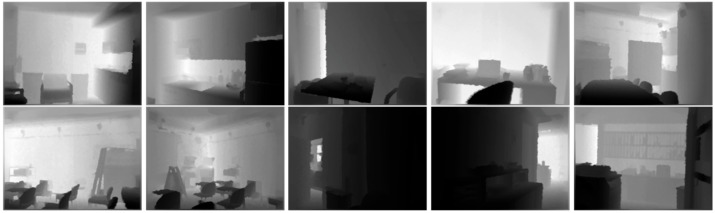
Samples of the dataset for the train network.

**Figure 11 sensors-22-09656-f011:**
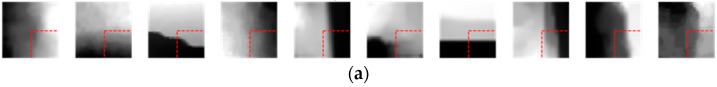
Cluster prediction results based on the loss function: (**a**) 32 × 32 input blocks; (**b**) 32 × 32 cluster probability maps with only MSE loss applied; (**c**) 32 × 32 cluster probability maps with the penalty added to the loss function. In (**b**,**c**), densities of red, green and blue means the probability for clusters.

**Figure 12 sensors-22-09656-f012:**
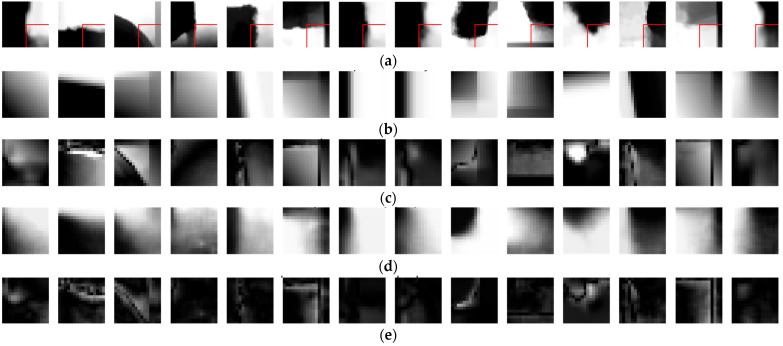
Comparison between the intra prediction performances of VVC and the proposed method: (**a**) 32 × 32 input blocks; (**b**) 16 × 16 error blocks of intra prediction by VVC; (**c**) 16 × 16 residual blocks of intra prediction by VVC; (**d**) 16 × 16 error blocks of intra prediction by proposed method; (**e**) 16 × 16 residual blocks of intra prediction by proposed method.

**Figure 13 sensors-22-09656-f013:**
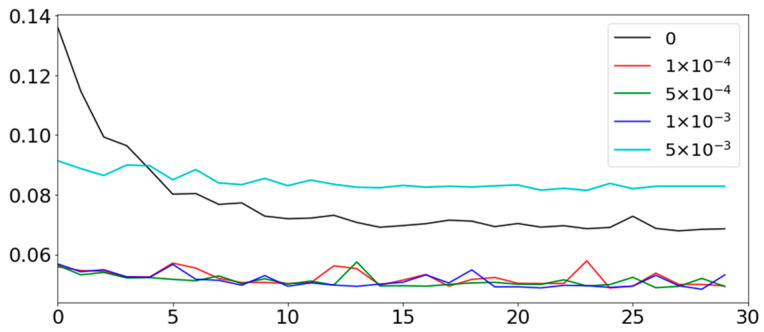
Intra prediction errors according to penalty weight in loss function.

**Figure 14 sensors-22-09656-f014:**
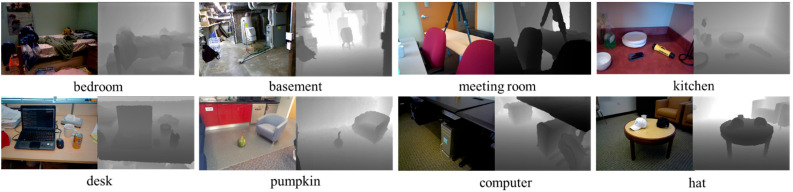
Depth videos for measuring depth video coding performance.

**Figure 15 sensors-22-09656-f015:**
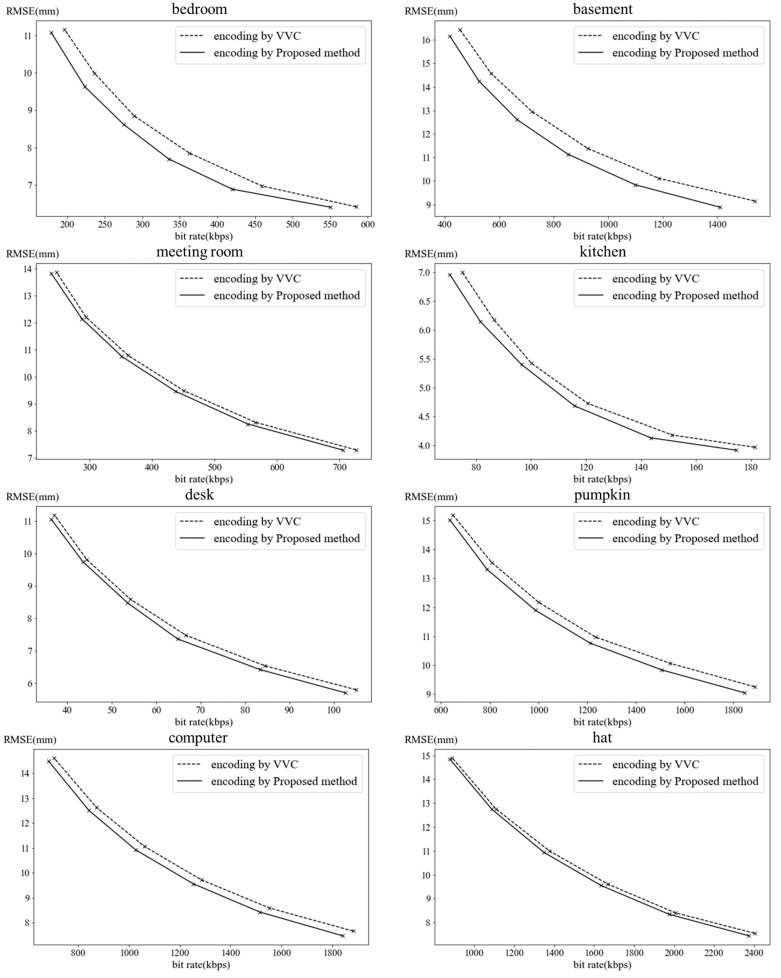
Comparison between rate-distortion curves for depth video coding through proposed method and VVC.

**Table 1 sensors-22-09656-t001:** Specifications of layers in the spatial feature prediction network.

No. Layer	From No.	Layer Name	No. Kernel	Shape of Kernel	Stride/ Padding	Activation Function	Output Shape
1		Input	-	-	-	-	(32,32,1)
2	1	Top block	-	-	-	-	(16,32,1)
3	1	Left block	-	-	-	-	(32,16,1)
4	3	Transpose	-	-	-	-	(16,32,1)
5	2 4	Conv_1	32	(16,32)	(1,1)/ (0,0)	PReLU	(1,30,32) × 2
6	5	Dense	1024	-	-	PReLU	-
7	6	Reshape	-	-	-	-	(32,32,1) × 2
8	7	Concatenate	-	-	-	-	(32,32,2)

**Table 2 sensors-22-09656-t002:** Description of layers in the clustering network.

No. Layer	From No.	Layer Name	No. Kernel	Shape of Kernel	Stride/ Padding	Activation Function	Output Shape
1		Input	-	-	-	-	(32,32,2)
2	1	Conv_1	12	(3,3)	(1,1)/ (2,2)	PReLU	(32,32,2)
3	1	Conv_2	6	(1,1)	(1,1)/ (0,0)	PReLU	(32,32,2)
4	3	Conv_3	6	(3,3)	(1,1)/ (2,2)	PReLU	(32,32,2)
8	7	Conv_4	*C*	(3,3)	(1,1)/ (2,2)	softmax	(32,32,*C*)

**Table 3 sensors-22-09656-t003:** Comparison of prediction errors based on number of clusters.

Number of Clusters	Prediction Error (MSE)
2	0.0804
3	0.0643
4	0.5426

**Table 4 sensors-22-09656-t004:** Prediction performances based on structure of spatial feature prediction network.

Depth of Fully Connected Layer	Prediction Error (MSE)	No. Parameters of FC Layer	No. Parameters of Whole Network	Processing Time (s)
1	0.0643	989,184	1,017,034	0.012
2	0.0629	2,043,904	2,071,754	0.016
4	0.0617	4,153,344	4,181,194	0.022

**Table 5 sensors-22-09656-t005:** Prediction performances based on structure of cluster prediction network.

Depth of CNN Layer	Prediction Error (MSE)	No. Parameters of CNN Layer	No. Parameters of Whole Network	Processing Time (s)
3	0.0709	18,848	1,010,560	0.012
4	0.0643	25,322	1,017,034	0.012
5	0.0637	31,796	1,023,508	0.012
6	0.0635	38,270	1,029,982	0.013

**Table 6 sensors-22-09656-t006:** Improvement of bit rate for depth video by proposed method compared with VVC.

Depth Video	RMSE	Intra Prediction Method	
		Plane Modeling [8]	Proposed Method
bedroom	10 mm	3.06%	8.00%
	15 mm	2.79%	5.74%
basement	10 mm	2.96%	12.45%
	15 mm	2.23%	10.63%
meeting room	10 mm	3.60%	4.55%
	15 mm	2.65%	2.64%
kitchen	10 mm	3.66%	3.10%
	15 mm	2.78%	1.78%
desk	10 mm	3.35%	4.31%
	15 mm	2.51%	3.19%
pumpkin	10 mm	5.83%	5.28%
	15 mm	5.65%	3.75%
computer	10 mm	6.76%	6.83%
	15 mm	6.66%	4.15%
hat	10 mm	4.06%	1.93%
	15 mm	3.76%	1.97%

**Table 7 sensors-22-09656-t007:** Distortion improvement for depth video by proposed method compared with VVC.

Depth Video	Bit Rate	Intra Prediction Method	
		Plane Modeling [8]	Proposed Method
bedroom	500 kbps	2.60%	6.07%
	1000 kbps	2.28%	4.13%
basement	500 kbps	1.05%	8.04%
	1000 kbps	0.90%	5.55%
meeting room	500 kbps	2.51%	3.36%
	1000 kbps	2.53%	3.95%
kitchen	500 kbps	12.07%	3.20%
	1000 kbps	12.70%	2.45%
desk	500 kbps	2.01%	3.50%
	1000 kbps	2.29%	3.43%
pumpkin	500 kbps	2.84%	2.14%
	1000 kbps	2.30%	2.38%
computer	500 kbps	4.16%	3.02%
	1000 kbps	3.15%	2.82%
hat	500 kbps	2.34%	1.81%
	1000 kbps	2.16%	1.03%

## Data Availability

Not applicable.

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
