# Peer review of "Intra Prediction Method for Depth Video Coding by Block Clustering through Deep Learning"

_sensors, 2022, doi:10.3390/s22249656_

Round 1

Reviewer 1 Report

This paper proposed the intra-picture prediction method for depth video by a block clustering through a neural network.

Authors used clustering by deep learning approach.  I have the following concerns to improve this manuscript:

- Abstract is able to give whole summarization of this manuscript. For the proposed algorithm, it would be better to describe more detailed fashion.

- I think key issue is clustering in this paper. So I suggest to add "clustering" in Keywords part.

- Authors mentioned "The previous studies of intra prediction by neural networks predict the spatial features through RNNs (Recurrent Neural Network) from the series of vertical or horizontal 

pixels [9-10]."  Is this true?  RNN is usually used to analyzed time series data. Intra prediction is focused on spatial feature.

- In Section 2.2, there are some recent works for prediction as:

  . CNN-based Fast Split Mode Decision Algorithm for Versatile Video Coding (VVC) Inter Prediction, Journal of Multimedia Information System (KMMS), vol. 8, no. 3, pp. 147-158 ( https://doi.org/10.33851/JMIS.2021.8.3.147 ), Sept. 2021

  . "CNN Based Optimal Intra Prediction Mode Estimation in Video Coding," 2020 IEEE International Conference on Consumer Electronics (ICCE), 2020, pp. 1-2,

  . CNN-based Approach for Visual Quality Improvement on HEVC, IEEE International Conference on Consumer Electronics (ICCE) (IEEE), pp. 498-500, Lasvegas USA, Jan. 1114, 2018

  It would better to add those in the above.

- In Fig. 2, authors showed the proposed structure and used 32x32 block size. As I know, there are sevral blcok sizes in video coding standard. If the size is different, how does it work?

- In Clustering layer, it was not clear to make what is outpu and what is the detailed role of this module. I think this part is major contribution, so we need more detailed description.

- In Figure 10, the sub-caption numbers are needed like (a) input blocks (b) ~~~~.

- The performance was shown well. I think it would be better to analyze the complexity in terms of No. of parameters and processing time.

- In Table 4 and 5, we can observe not better BD rate in some sequence such as Kitchen. Why this happens? It should be explained in more detail.

- There are some typos as:

  In p. 4, 

    . "Figure 1. Input block for proposed network." --> "Figure 1. Input block for the proposed network." 

    . "Figure 2. Structure of proposed network. " --> "Figure 2. Structure of the proposed network. "

  Authors should polish them very carefully on the whole manuscript.

Author Response

We would like to express our appreciations for your review. We believe that your detailed comments and suggestions have contributed substantially to improve the presentation of our study, as well as its overall quality and the manuscript. Following, we offer replies to the points the reviewer addressed regarding the original manuscript.

Point 1: Abstract is able to give whole summarization of this manuscript. For the proposed algorithm, it would be better to describe more detailed fashion.

Response 1: We described the scheme of the proposed network and the intra prediction algorithm to the abstract in more detail as follows.

  • In this paper, we propose an intra-picture prediction method for depth video by a block clustering through a neural network. The proposed method solves a problem that the block that has two or more clusters drops the prediction performance of the intra prediction for depth video. The proposed neural network consists of both a spatial feature prediction network and a clustering network. The spatial feature prediction network utilizes spatial features in vertical and horizontal directions. The network contains a 1D CNN layer and a fully connected layer. The 1D CNN layer extracts the spatial features for a vertical direction and a horizontal direction from a top block and a left block of the reference pixels, respectively. 1D CNN is designed to handle time-series data, but it can also be applied to find the spatial features by regarding a pixel order in a certain direction as a timestamp. The fully connected layer predicts the spatial features of the block to be coded through the extracted features. The clustering network finds clusters from the spatial features which are the outputs of the spatial feature prediction network. The network consists of 4 CNN layers. The first 3 CNN layers combine two spatial features in the vertical and horizontal directions. The last layer outputs the probabilities that pixels belong to the clusters. The pixels of the block are predicted by the representative values of the clusters that are the average of the reference pixels belonging to the clusters. For the intra prediction for various block sizes, the block is scaled to the size of the network input. The prediction result through the proposed network is scaled back to the original size. In network training, the mean square error is used as a loss function between the original block and the predicted block. A penalty for output values far from both ends is introduced to the loss function for clear network clustering. In the simulation results, the bit rate is saved by up to 12.45% under the same distortion condition compared with the latest video coding standard.

Point 2: I think key issue is clustering in this paper. So I suggest to add "clustering" in Keywords part.

 Response 2: We added a ‘clustering’ in the Keywords part.

Point 3: Authors mentioned "The previous studies of intra prediction by neural networks predict the spatial features through RNNs (Recurrent Neural Network) from the series of vertical or horizontal pixels [9-10]."  Is this true?  RNN is usually used to analysed time series data. Intra prediction is focused on spatial feature.

 Response 3: We described the method of the intra prediction through RNN to new section ‘3.1. Spatial Feature Extraction through 1D CNN’ as follows.

  • (Line 164) In conventional video coding standards such as HEVC and VVC, the spatial features of the block are extracted through the various modes of angular intra predictions. The an-gular mode predicts pixels as a reference pixel in a certain direction. Figure 1a shows the predictions results of the angular modes in a vertical and horizontal directions. The errors of the predictions are calculated for the angular intra prediction mode and a DC mode, which predicts pixels as the average of the reference pixels. Then, the prediction mode is selected to make the prediction error smallest. The angular intra prediction can extract the spatial features in the linear direction well, but non-linear spatial features can hardly be extracted as shown in Figure 1c.

    Though RNN is designed for the continuous data in essence, RNN can extract the spatial features of a video if a pixel order in a certain direction is regarded as a timestamp. RNN extracts the features based on the data order in 1D domain, but the spatial features of the video are in 2D domain. Therefore, it is necessary to appropriately combine the spatial features in various directions in order to extract the spatial features in the 2D domain. In PS-RNN [10], RNN layers extract two spatial features in the vertical and the horizontal directions. CNN layers combine both of the spatial features and predict the block. Figure 2a shows the extraction of the spatial features through RNN. The spatial feature extraction through RNN can extract non-linear spatial features that conventional angular modes cannot. 1D CNN can replace RNN in the spatial feature extraction of the video similar to the feature extraction of the time series data. 1D CNN can avoid gradient vanishing or exploding in network training, which is the critical problem of RNN. 1D CNN has equal performance to or better performance than RNN [11-12]. The kernel moves the input block, which is the top or the left reference pixels, in one direction and extracts the vertical or horizontal spatial features, respectively. Figure 2b shows the extraction of the vertical spatial features through 1D CNN.

Point 4: In Section 2.2, there are some recent works for prediction as

  • CNN-based Fast Split Mode Decision Algorithm for Versatile Video Coding (VVC) Inter Prediction, Journal of Multimedia Information System (KMMS), vol. 8, no. 3, pp. 147-158 ( https://doi.org/10.33851/JMIS.2021.8.3.147 ), Sept. 2021
  • "CNN Based Optimal Intra Prediction Mode Estimation in Video Coding," 2020 IEEE International Conference on Consumer Electronics (ICCE), 2020, pp. 1-2,
  • CNN-based Approach for Visual Quality Improvement on HEVC, IEEE International Conference on Consumer Electronics (ICCE) (IEEE), pp. 498-500, Lasvegas USA, Jan. 1114, 2018

  It would better to add those in the above.

Response 4:  We added these works in Related Works section and cited them as follows.

  • (Line 120) The neural networks with CNN can improve computational complexities of the block split [34] and the mode selection [35] for the intra prediction. proposes the video quality enhancement method in video decoding by the picture prediction through CNN. Lee et al. [36] propose the video quality enhancement method in video decoding by the picture prediction through CNN.

Point 5: In Fig. 2, authors showed the proposed structure and used 32x32 block size. As I know, there are several block sizes in video coding standard. If the size is different, how does it work?.

 Response 5: Figure 2 shows the input and output in the block clustering network. The flow of actual intra prediction is shown in Figure 9. The blocks to be coded are scaled to the size of the network input through interpolation. The predicted result block through the network is scaled back to its original size. It is described in Section 3.3. We added the description of the intra prediction for the various sized block to Abstract.

Point 6: In Clustering layer, it was not clear to make what is output and what is the detailed role of this module. I think this part is major contribution, so we need more detailed description.

 Response 6: We added the role and the output of the clustering network as follows.

  • (Line 244) The first CNN layer extracts 12 features combining vertical and horizontal directions. The second CNN layer reduces the dimension of the feature map by a 1×1 kernel. The third CNN layer refines the spatial features. The cluster probability for each pixel is calculated by the softmax activation function in the last CNN layer. The output of the clustering net-work is a 32×32×C cluster probability map. The network outputs are cluster probabilities for the input block. The ranges of the output values are [0, 1]. What an output value in (i, j, c) position is closer to 1 means a pixel in (i, j) position has higher probability to belong to cth cluster.

Point 7: In Figure 10, the sub-caption numbers are needed like (a) input blocks (b) .

 Response 7: We modified the captions of Figure 10 by sub-numbering as follows.

  • Figure 12. Comparison between the intra prediction performances of VVC and the proposed method: (a) 32×32 input blocks; (b) 16×16 error blocks of intra prediction by VVC; (c) 16×16 residual blocks of intra prediction by VVC; (d) 16×16 error blocks of intra prediction by proposed method; (e) 16×16 residual blocks of intra prediction by proposed method.

Point 8: The performance was shown well. I think it would be better to analyze the complexity in terms of No. of parameters and processing time.

 Response 8: We analysed the complexity of the network and measured the processing times as follows.

  • (Line 349) We also measure the complexities and the processing time of the network. The complexity of the network is measured as the number of the operations in the network, which is the number of the network parameters. The processing time is averaged by measuring the processing time for a block 100 times. Even though the intra prediction performance improves by adding the fully connected layer, the complexity of the network grows much faster than the increase of the prediction accuracy.
  • (Line 361) The complexity of the network and the processing time less increases then the case of in-creases of the fully connected layers.

Point 9: In Table 4 and 5, we can observe not better BD rate in some sequence such as Kitchen. Why this happens? It should be explained in more detail.

Response 9: We analysed why the previous method performed better for some depth videos as follows.

  • (Line 408) The previous method is better than the proposed method for coding the depth videos of hat and kitchen, which has few objects with simple background. The previous method performs the surface-based prediction, so then it greatly improves the bit rates and the distortions for the depth videos with a simple background and little number of objects. On the other hand, the proposed method improves the coding performance for the depth videos with a complex background and many objects.

Point 10: There are some typos. Authors should polish them very carefully on the whole manuscript.

Response 10: We corrected typos in the manuscript.

We would like to appreciate your detailed review again.

Reviewer 2 Report

This paper deals with the compression of distance-related data through block-based predictions. Currently, the use of depth data is increasing in many fields, such as object detection for autonomous driving. This paper proposes a depth data compression method that predicts the spatial features in vertical and horizontal directions by a CNN network.

To improve this paper, my comments are as the followings:

(1)Abstract should explain more details of the proposed scheme.

(2)To address the contributions, the authors should compare and explain the differences between the SOTA depth in-screen prediction methods and the proposed method.

(3)In the proposed scheme, the authors used 1D CNN to predict spatial features. Actually spatial feature is usually from 2D texture. Thus the authors should explain why 1D CNN is more effective in detail.

(4)It is necessary to further explain the process of converting 3D coordinates (X, Y, Z) into the expressions related to pixel and image coordinates in Eq. (7).

(5)A more detailed description of the network structure is needed. For example, the authors can plot the composition of each layer of the spatial feature prediction layer.

(6)The writing should be improved by carefully checking on the expression mistakes and some errors.

Author Response

We would like to express our appreciations for your review. We believe that your detailed comments and suggestions have contributed substantially to improve the presentation of our study, as well as its overall quality and the manuscript. Following, we offer replies to the points the reviewer addressed regarding the original manuscript.

Point 1: Abstract should explain more details of the proposed scheme.

Response 1: We described the scheme of the proposed network in detail as follows.

  • In this paper, we propose an intra-picture prediction method for depth video by a block clustering through a neural network. The proposed method solves a problem that the block that has two or more clusters drops the prediction performance of the intra prediction for depth video. The proposed neural network consists of both a spatial feature prediction network and a clustering network. The spatial feature prediction network utilizes spatial features in vertical and horizontal directions. The network contains a 1D CNN layer and a fully connected layer. The 1D CNN layer extracts the spatial features for a vertical direction and a horizontal direction from a top block and a left block of the reference pixels, respectively. 1D CNN is designed to handle time-series data, but it can also be applied to find the spatial features by regarding a pixel order in a certain direction as a timestamp. The fully connected layer predicts the spatial features of the block to be coded through the extracted features. The clustering network finds clusters from the spatial features which are the outputs of the spatial feature prediction network. The network consists of 4 CNN layers. The first 3 CNN layers combine two spatial features in the vertical and horizontal directions. The last layer outputs the probabilities that pixels belong to the clusters. The pixels of the block are predicted by the representative values of the clusters that are the average of the reference pixels belonging to the clusters. For the intra prediction for various block sizes, the block is scaled to the size of the network input. The prediction result through the proposed network is scaled back to the original size. In network training, the mean square error is used as a loss function between the original block and the predicted block. A penalty for output values far from both ends is introduced to the loss function for clear network clustering. In the simulation results, the bit rate is saved by up to 12.45% under the same distortion condition compared with the latest video coding standard.

Point 2: To address the contributions, the authors should compare and explain the differences between the SOTA depth in-screen prediction methods and the proposed method.

 Response 2: We explained the state-of-the-art methods of depth picture prediction in Section 2.2 as follows.

  • (Line 126) Depth video has three-dimension spatial information. Therefore, depth video can be converted to a point cloud which is a discrete set of three-dimensional points. MPEG standardizes the point cloud compression as V-PCC (Video-based Point Cloud Compression) [28], which is a patch-based cloud point compression method. After splitting the 3D points in the point cloud into patches with high spatial correlation, each patch is projected onto a 2D surface. The projected patches are compressed through the prediction methods in the conventional video coding standards such as HEVC and VVC. However, the point cloud compression method by 2D projection removes the spatial correlation of Z-axis, so the compression performance has a limit. The point cloud can also be compressed through 3D spatial correlation. Many studies of the point cloud compression generate an octree for dividing 3D space. The octree is a tree structure where nodes represent bounding boxes that are recursively divided into eight leaves. Similar to divided 2D image into blocks through a quadtree, the 3D space can be divided into sub-cubes through octree. The loss and the rate of the compression can be determined by adjusting the depth of the octree. Garcia [29] proposes a method of compressing the flags of leaf nodes and their parents in the octree through LZW algorithm and arithmetic coding. Kathariya [30] proposes a BTQT (Binary Tree Quadtree) structure for compressing the point cloud. The points in the point cloud are split into two sets through a binary tree. A set which presents a plane surface is compressed by converting it into a quadtree. The other is compressed through the octree. The point cloud can also be compressed through voxelization [31]. Adjacent 3D points are converted into a single voxel through voxelization. The point cloud compression methods through 3D spatial correlation are more precise predictions than 2D projection methods. However, these methods have a limitation that has a high computational complexity due to calculating the 3D spatial correlation.

  Depth video can also be treated as a single channel video whose range of pixel representation is extended. Therefore, depth video can be compressed by the conventional video coding standards. Nenci [32] proposes a depth video coding method through depth video conversion. The channel of depth video is divided into 8-bit multi-channels. The multi-channel video is compressed through AVC. Wang [33] proposed the inter-frame prediction method of finding camera movements between temporally adjacent depth pictures. Our previous study [8] proposes a new intra prediction mode based on a plane surface estimation. The 3D plane surface is estimated through the reference pixels. The depth pixels are predicted through the estimated plane surface. The study greatly improves the performance especially of intra prediction for the depth pictures with simple backgrounds.

Point 3: In the proposed scheme, the authors used 1D CNN to predict spatial features. Actually, spatial feature is usually from 2D texture. Thus, the authors should explain why 1D CNN is more effective in detail.

 Response 3: We described the method of the intra prediction through RNN to new section ‘3.1. Spatial Feature Extraction through 1D CNN’ as follows.

  • (Line 164) In conventional video coding standards such as HEVC and VVC, the spatial features of the block are extracted through the various modes of angular intra predictions. The an-gular mode predicts pixels as a reference pixel in a certain direction. Figure 1a shows the predictions results of the angular modes in a vertical and horizontal directions. The errors of the predictions are calculated for the angular intra prediction mode and a DC mode, which predicts pixels as the average of the reference pixels. Then, the prediction mode is selected to make the prediction error smallest. The angular intra prediction can extract the spatial features in the linear direction well, but non-linear spatial features can hardly be extracted as shown in Figure 1c.

  Though RNN is designed for the continuous data in essence, RNN can extract the spatial features of a video if a pixel order in a certain direction is regarded as a timestamp. RNN extracts the features based on the data order in 1D domain, but the spatial features of the video are in 2D domain. Therefore, it is necessary to appropriately combine the spatial features in various directions in order to extract the spatial features in the 2D domain. In PS-RNN [10], RNN layers extract two spatial features in the vertical and the horizontal directions. CNN layers combine both of the spatial features and predict the block. Figure 2a shows the extraction of the spatial features through RNN. The spatial feature extraction through RNN can extract non-linear spatial features that conventional angular modes cannot. 1D CNN can replace RNN in the spatial feature extraction of the video similar to the feature extraction of the time series data. 1D CNN can avoid gradient vanishing or exploding in network training, which is the critical problem of RNN. 1D CNN has equal performance to or better performance than RNN [11-12]. The kernel moves the input block, which is the top or the left reference pixels, in one direction and extracts the vertical or horizontal spatial features, respectively. Figure 2b shows the extraction of the vertical spatial features through 1D CNN.

Point 4: It is necessary to further explain the process of converting 3D coordinates (X, Y, Z) into the expressions related to pixel and image coordinates in Eq. (7).

 Response 4: We Added the conversion of 2D image coordinates (i, j)to 3D coordinates (X, Y, Z)  as follows.

  • (Line 385) The depth value means a Z-axis coordinate in a 3D coordinate system, so the 2D coordinates (i, j) with a depth pixel value p(i, j) can be converted to the 3D coordinates as follows,

X=i / f * p( i ,j )

Y=j / f * p( i, j )

Z=p( i, j )

where f is the focus length of the depth camera.

Point 5: A more detailed description of the network structure is needed. For example, the authors can plot the composition of each layer of the spatial feature prediction layer.

 Response 5: We described the specifications of the spatial feature prediction network and the clustering network as follows.

Table 1. Specifications of layers in the spatial feature prediction network.

No. layer

From No.

Layer name

No. kernel

Shape of kernel

Stride/
padding

Activation function

Output shape

1

Input

-

-

-

-

(32,32,1)

2

1

Top block

-

-

-

-

(16,32,1)

3

1

Left block

-

-

-

-

(32,16,1)

4

3

Transpose

-

-

-

-

(16,32,1)

5

2

4

Conv_1

32

(16,32)

S:(1,1)
P:(0,0)

PReLU

(1,30,32)×2

6

5

Dense

1024

-

-

PReLU

-

7

6

Reshape

-

-

-

-

(32,32,1)×2

8

7

Concatenate

-

-

-

-

(32,32,2)

* S and P mean the sizes of a stride and a padding, respectively.

Table 2. Description of layers in the clustering network.

No. layer

From No.

Layer name

No. kernel

Shape of kernel

Stride/
padding

Activation function

Output shape

1

Input

-

-

-

-

(32,32,2)

2

1

Conv_1

12

(3,3)

S:(1,1)
P:(2,2)

PReLU

(32,32,2)

3

1

Conv_2

6

(1,1)

S:(1,1)
P:(0,0)

PReLU

(32,32,2)

4

3

Conv_3

6

(3,3)

S:(1,1)
P:(2,2)

PReLU

(32,32,2)

8

7

Conv_4

C

(3,3)

S:(1,1)
P:(2,2)

softmax

(32,32,C)

* S and P mean the sizes of a stride and a padding, respectively. C is the number of the clusters.

Point 6: The writing should be improved by carefully checking on the expression mistakes and some errors.

Response 6: We corrected the expression mistakes and errors in the manuscript.

We would like to appreciate your detailed review again.

Round 2

Reviewer 2 Report

Accept.